# Immunonutrition, Metabolism, and Programmed Cell Death in Lung Cancer: Translating Bench to Bedside

**DOI:** 10.3390/biology13060409

**Published:** 2024-06-04

**Authors:** Palma Fedele, Anna Natalizia Santoro, Francesca Pini, Marcello Pellegrino, Giuseppe Polito, Maria Chiara De Luca, Antonietta Pignatelli, Michele Tancredi, Valeria Lagattolla, Alessandro Anglani, Chiara Guarini, Antonello Pinto, Pietro Bracciale

**Affiliations:** 1Oncology Unit, Dario Camberlingo Hospital, 72021 Francavilla Fontana, Italy; annanatalizia.santoro@asl.brindisi.it (A.N.S.); fr.chicca@gmail.com (F.P.); antonello.pinto@studenti.unimi.it (A.P.); 2Pathology Unit, Antonio Perrino Hospital, 72100 Brindisi, Italy; marcello.pellegrino@asl.brindisi.it; 3Nuclear Medicine Unit, Antonio Perrino Hospital, 72100 Brindisi, Italy; giuseppe.polito@asl.brindisi.it; 4Radiotherapy Unit, Antonio Perrino Hospital, 72100 Brindisi, Italy; chiara.deluca@asl.brindisi.it; 5Palliative Care Unit, Antonio Perrino Hospital, 72100 Brindisi, Italy; antonietta.pignatelli@asl.brindisi.it; 6Radiology Unit, Antonio Perrino Hospital, 72100 Brindisi, Italy; michele.tancredi@asl.brindisi.it; 7Clinic Nutrition Unit, Antonio Perrino Hospital, 72100 Brindisi, Italy; valeria.lagattolla@asl.brindisi.it; 8Radiology Unit, Dario Camberlingo Hospital, 72021 Francavilla Fontana, Italy; alessandro.anglani@asl.brindisi.it; 9Course in Development and Production of Biotechnological Drugs, Faculty of Pharmaceutical Science, University of Milan, 20122 Milano, Italy; 10Pneumology Unit, Ostuni Hospital, 72017 Ostuni, Italy; pietro.bracciale@asl.brindisi.it

**Keywords:** immunonutrition, lung cancer, apoptosis, metabolism, programmed cell death

## Abstract

**Simple Summary:**

Lung cancer presents significant therapeutic challenges, driving the exploration of novel treatment strategies. Programmed cell death (PCD) mechanisms, including apoptosis, autophagy, and programmed necrosis, are crucial in lung cancer pathogenesis and treatment response. Dysregulation of these pathways contributes to tumor progression and therapy resistance. Immunonutrition, using specific nutrients to modulate immune function, and metabolic reprogramming, a hallmark of cancer cells, offer promising intervention avenues. Nutritional interventions, such as omega-3 fatty acids, modulate PCD pathways in cancer cells, while targeting metabolic pathways implicated in apoptosis regulation presents a compelling therapeutic approach. Clinical evidence supports the role of immunonutritional interventions, including omega-3 fatty acids, in enhancing PCD and improving treatment outcomes in lung cancer patients. Additionally, synthetic analogues of natural compounds, like resveratrol, show promising anticancer properties by modulating apoptotic signaling pathways. This review highlights the convergence of immunonutrition, metabolism, and PCD pathways in lung cancer biology, emphasizing the potential for therapeutic exploration. Further elucidation of the specific molecular mechanisms governing these interactions is crucial for translating these findings into clinical practice and enhancing lung cancer management.

**Abstract:**

Lung cancer presents significant therapeutic challenges, motivating the exploration of novel treatment strategies. Programmed cell death (PCD) mechanisms, encompassing apoptosis, autophagy, and programmed necrosis, are pivotal in lung cancer pathogenesis and the treatment response. Dysregulation of these pathways contributes to tumor progression and therapy resistance. Immunonutrition, employing specific nutrients to modulate immune function, and metabolic reprogramming, a hallmark of cancer cells, offer promising avenues for intervention. Nutritional interventions, such as omega-3 fatty acids, exert modulatory effects on PCD pathways in cancer cells, while targeting metabolic pathways implicated in apoptosis regulation represents a compelling therapeutic approach. Clinical evidence supports the role of immunonutritional interventions, including omega-3 fatty acids, in augmenting PCD and enhancing treatment outcomes in patients with lung cancer. Furthermore, synthetic analogs of natural compounds, such as resveratrol, demonstrate promising anticancer properties by modulating apoptotic signaling pathways. This review underscores the convergence of immunonutrition, metabolism, and PCD pathways in lung cancer biology, emphasizing the potential for therapeutic exploration in this complex disease. Further elucidation of the specific molecular mechanisms governing these interactions is imperative for translating these findings into clinical practice and improving lung cancer management.

## 1. Introduction

Lung cancer remains a significant public health challenge worldwide, with high mortality rates and limited treatment options, particularly in the advanced stages of the disease. Despite advances in conventional therapies, including surgery, chemotherapy, immunotherapy, targeted therapy, and radiation therapy [1], the prognosis for many lung cancer patients remains poor. Therefore, there is a growing interest in exploring alternative treatment approaches that target specific molecular pathways involved in cancer development and progression [2]. One such promising avenue is the manipulation of programmed cell death (PCD) pathways, particularly apoptosis, which plays a crucial role in regulating cell proliferation and survival [3]. Apoptosis is a tightly regulated process that allows multicellular organisms to eliminate damaged or abnormal cells, and its dysregulation has been implicated in various diseases, including cancer [4]. In the context of lung cancer, understanding the interplay between apoptosis and other cellular processes, such as immunonutrition and metabolism, holds significant therapeutic potential [5]. Immunonutrition refers to the use of specific nutrients and bioactive compounds to modulate immune function and enhance the body’s response to disease [6]. Mounting evidence suggests that immunonutritional interventions can influence PCD pathways in cancer cells, offering a novel approach to cancer therapy [7]. Additionally, metabolic reprogramming is a hallmark of cancer cells, and targeting metabolic pathways involved in apoptosis regulation may provide new opportunities for treatment [8]. This review aims to explore the intersection of immunonutrition, PCD, metabolism, and lung cancer. We will examine the molecular mechanisms underlying PCD pathways [9], discuss the impact of immunonutritional interventions on apoptosis in lung cancer cells [10], and explore the potential of metabolic targeting for enhancing PCD in lung cancer [11]. Furthermore, we will discuss the clinical implications of these findings and the potential translation of bench research into bedside applications for patients with lung cancer [12].

## 2. Spectrum of Programmed Cell Death Modalities

PCD is crucial for maintaining the delicate balance between cell survival and death in healthy cells, yet its disruption is pivotal in determining the destiny of cancer cells [13,14]. PCD encompasses three primary types including apoptosis (type I) [15], autophagy (type II) [16], and programmed necrosis (type III) [17], each distinguished by unique morphological characteristics. Apoptosis involves cellular contraction, nuclear condensation, membrane bulging, and biochemical changes such as DNA cleavage and phosphatidylserine externalization. Apoptosis occurs through two pathways including the extrinsic pathway and the intrinsic pathway. The extrinsic pathway is triggered by the binding of death ligands such as TNFα, FasL, and TRAIL to their respective cell surface receptors, leading to the activation of caspase-8 and the initiation of apoptosis.

The intrinsic pathway is activated by irreparable damage to cellular components and is regulated by Bcl-2 family proteins. This pathway involves the release of cytochrome C and SMAC/DIABLO (second mitochondria-derivated activator of caspase/direct inhibitor of apoptosis- binding protein with low pi) from mitochondria, leading to the activation of caspase 9 and the induction of apoptosis in cancer cells [5].

Autophagy begins with the formation of self-digesting structures known as autophagosomes, which engulf cellular components for recycling, and it can either support cell survival during stress or induce cell death under extreme conditions. Autophagy plays a dual role in regulating various physiological processes including starvation, cellular survival, and differentiation [16]. Programmed necrosis entails cellular swelling, organelle dysfunction, and dissolution, contributing to tissue balance and the removal of compromised cells, which has significant implications for cancer tissues [17].

## 3. Mechanisms and Modalities of Programmed Cell Death in Cancer

PCD mechanisms are intricately involved in the relationship with cancer, encompassing several pathways [18]. Apoptosis, triggered by both extrinsic and intrinsic pathways, involves the activation of caspases, leading to cell death when DNA damage is irreparable. The mechanisms of apoptosis are finely regulated. Disruption of the balance between survival and death, often caused by alterations in the expression or function of key proteins involved in apoptotic signaling pathways, can lead to enhanced cellular survival, thereby promoting the development and progression of cancer. One of the central players in apoptosis regulation is the tumor suppressor protein p53. p53 acts as a guardian of the genome by responding to various stress stimuli, such as DNA damage and oncogene overexpression, and inducing either DNA repair or apoptosis. Post-translational modifications, including phosphorylation, acetylation, and ubiquitination, regulate the activation of p53 in response to cellular stress. The oncoprotein MDM2 serves as a critical negative regulator of p53. MDM2 inhibits the transcriptional activity of p53 and promotes its degradation through ubiquitination, thereby controlling cellular responses to stress and maintaining tissue homeostasis. Dysregulation of the p53-MDM2 axis, often observed in cancer, contributes to tumor development by allowing aberrant cellular proliferation and survival. p53 exerts its tumor suppressive function through multiple mechanisms, including the transcriptional activation of pro-apoptotic genes. p53 induces the expression of pro-apoptotic Bcl-2 family proteins, such as Bax, Puma, Noxa, and Bid, promoting the intrinsic apoptotic pathway involving mitochondrial dysfunction. Additionally, p53 regulates the expression of death receptor genes like DR4, DR5, and Fas, activating the extrinsic apoptotic pathway. In addition to its transcriptional regulation, p53 can induce apoptosis through transcription-independent mechanisms. p53 directly interacts with Bcl-2 family proteins, leading to the activation of the intrinsic apoptotic pathway. Mitochondrial translocation of p53 facilitates interactions with both anti-apoptotic and pro-apoptotic Bcl-2 family members, ultimately promoting apoptosis. The inactivation of p53, commonly observed in human cancers through gene mutations or alterations in its regulators, contributes to tumor development and progression. Loss of functional p53 facilitates cellular transformation by promoting cell survival and the persistence of genetic defects. Efforts are underway to develop therapeutic strategies that target the restoration of wild-type p53 function in cancer cells. These approaches include gene transfer of wild-type p53, inhibition of MDM2-p53 interaction, and chemical restoration of p53 activity. Restoring p53 function in cancer cells aims to counteract their aggressive behavior and enhance their susceptibility to anti-cancer therapies. The Jun N-terminal kinase (JNK) pathway also plays a significant role in apoptosis modulation. JNK responds to cellular stressors and regulates apoptosis through both pro- and anti-apoptotic effects. Activation of JNK promotes the expression of pro-apoptotic genes and targets anti-apoptotic Bcl-2 family proteins, enhancing apoptosis in cancer cells.

Autophagy, regulated by autophagy-related genes, can promote cell survival under stress but may also lead to autophagic cell death when over-activated, playing a dual role in cancer progression. The dual role of autophagy in tumor progression depends on factors such as tumor subtype and mutation status. In the precancerous stage, inhibition of autophagy can lead to the accumulation of reactive oxygen species and genomic dysfunction, ultimately promoting tumor formation. However, autophagy can also support tumor survival by providing energy and nutrients when stimulated by conditions like starvation or oxidative stress. Autophagy is regulated by autophagy-related genes such as ULK1, Beclin-1, LC3, p62, and FoxO, with ULK1 acting as a promoter of autophagy. Additionally, autophagy-associated signaling pathways, including PI3KC1-Akt-mTORC1, Ras-Raf-MAPKs, and NF-κB pathways, play crucial roles against tumor progression and metastasis.

Programmed necrosis, mediated by key factors like receptor-interacting serine/threonine protein kinases and poly (ADP-ribose) polymerase, has been recognized as a controlled form of cell death, potentially impacting tumor growth and inflammation. The interplay among apoptosis, autophagy, and programmed necrosis is complex, with synergistic or antagonistic effects depending on the cellular context, stage of carcinogenesis, and therapeutic interventions, highlighting the intricate balance between cell survival and death in cancer progression [19].

## 4. Role of Programmed Cell Death Mechanisms in Lung Cancer Pathogenesis

PCD mechanisms play pivotal roles in the pathogenesis of lung cancer. Dysregulation of these pathways contributes to tumor initiation, progression, and therapy resistance. Apoptosis is disrupted in lung cancer, leading to uncontrolled cell proliferation and survival. Dysregulation of apoptotic regulators such as p53, Bcl-2 family proteins, caspases, and inhibitors of apoptosis proteins has been implicated in lung cancer progression [2]. Studies have shown aberrant expression of anti-apoptotic proteins like Bcl-2 and Bcl-xL in lung cancer, conferring resistance to apoptosis-inducing therapies [20]. Targeting apoptotic pathways, such as through BH3 mimetics or caspase activators, represents a promising therapeutic strategy in lung cancer treatment [21].

Autophagy exhibits dual roles in lung cancer, acting as both a tumor suppressor and a promoter of tumorigenesis [22]. Dysregulated autophagy contributes to lung cancer progression by promoting cell survival under stress conditions and facilitating tumor growth [23]. The interplay between autophagy and apoptosis in lung cancer cells is complex, with studies suggesting both synergistic and antagonistic interactions [24]. Modulating autophagic flux using pharmacological agents or genetic manipulation presents an attractive avenue for therapeutic intervention in lung cancer [25]. Necroptosis, a programmed form of necrotic cell death, has garnered attention in the context of lung cancer, although its role is less well understood compared with apoptosis and autophagy [26]. Emerging evidence suggests a potential involvement of necroptosis in lung cancer pathogenesis, with studies implicating key regulators such as receptor-interacting protein kinase 3 (RIPK3) [27]. Further research is warranted to elucidate the precise contribution of necroptosis to lung cancer development and its potential as a therapeutic target [28].

## 5. The Interplay between Programmed Cell Death and Immunonutrition and Metabolism in Lung Cancer

Immunonutrition, defined as the use of specific nutrients to modulate immune function, and metabolism, which encompasses the complex network of biochemical processes governing energy production and macromolecule synthesis, are interconnected cellular processes that intersect with PCD pathways in cancer cells. While specific studies examining the direct interplay among immunonutrition, metabolism, and PCD in the context of lung cancer may be limited, there is growing evidence of their individual and collective impact on cancer biology [29,30,31]. Nutritional interventions targeting immunonutrition, such as omega-3 fatty acids, vitamins, minerals, and phytochemicals, have been investigated for their potential to modulate apoptosis and other forms of PCD in cancer cells [32,33,34].

Immunonutrition, metabolism, and PCD are interconnected cellular processes that influence each other and converge on the regulation of programmed cell death pathways. Immunonutrients can impact cellular metabolism by serving as substrates or cofactors for metabolic reactions, while metabolic intermediates can modulate immune cell function and inflammatory signaling. This cross-talk among immunonutrition, metabolism, and PCD underscores the complexity of cancer biology and highlights the potential synergistic effects of combining immunotherapy with metabolic interventions for cancer treatment.

Similarly, dysregulated metabolism, characterized by alterations in glycolysis, nutrient sensing pathways, and mitochondrial metabolism, can influence apoptotic signaling pathways in cancer [35,36,37]. Metabolic intermediates, including ATP, NAD+, and reactive oxygen species, play critical roles in regulating mitochondrial function, redox balance, and apoptotic protein activity [38,39]. The interconnection among immunonutrition, metabolism, and PCD in cancer underscores the complexity of cancer biology and the importance of considering multiple cellular processes in the development of therapeutic strategies [40].

Targeting metabolic pathways implicated in apoptosis and cancer represents a promising therapeutic avenue in cancer management, including lung cancer. Dysregulated metabolism stands as a hallmark of lung cancer, significantly contributing to uncontrolled cell proliferation, survival, and resistance to apoptosis [41]. Metabolic reprogramming within lung cancer cells encompasses alterations across various metabolic pathways including glucose metabolism, lipid metabolism, and amino acid metabolism [42]. The Warburg effect, characterized by enhanced glycolysis even under normoxic conditions, underscores one of the well-documented metabolic adaptations in lung cancer cells, providing them with a robust source of energy and biosynthetic precursors essential for sustaining rapid proliferation [43]. Furthermore, aberrant lipid metabolism and amino acid metabolism play pivotal roles in fueling lung tumor growth and survival by providing crucial building blocks for membrane synthesis and protein biosynthesis, respectively [44]. Addressing key metabolic pathways involved in lung cancer metabolism, such as glycolysis, fatty acid synthesis and oxidation, and amino acid metabolism, has emerged as a compelling therapeutic strategy [45]. Small molecule inhibitors selectively targeting enzymes crucial in these pathways, including hexokinase, lactate dehydrogenase, fatty acid synthase, and glutaminase, have exhibited promising efficacy in preclinical studies and are currently undergoing clinical investigation in patients with lung cancer [46]. By disrupting lung cancer cell metabolism, these targeted therapies aim to induce apoptosis and inhibit tumor growth, presenting a novel therapeutic approach for patients with lung cancer [47].

Patient variability, including genetic differences and lifestyle factors, affects the treatment response, indicating that personalized approaches are required. Treatment resistance can emerge because of tumor adaptations, necessitating the development of combination therapies targeting multiple vulnerabilities. Potential side effects, such as gastrointestinal disturbances and immune-related adverse events, need careful management to maintain treatment adherence. Integrating these interventions with standard therapies poses challenges in timing and sequencing to optimize outcomes. Clinical trial design must consider appropriate endpoints and recruitment of diverse patient populations to generate robust evidence. Despite challenges, understanding these limitations enables the optimization of therapeutic strategies for lung cancer.

## 6. Clinical Application of Metabolism and Immunonutrition in Targeting Programmed Cell Death in Lung Cancer

Recent studies have highlighted the interplay between metabolism and programmed cell death regulation in lung cancer. For instance, dysregulated glucose metabolism, characterized by increased glucose uptake and glycolysis, contributes to cancer cell survival and resistance to apoptosis [48,49]. Further investigations have highlighted the significant interplay among circadian rhythm disruption, metabolism alterations, and lung tumorigenesis. Utilizing a genetically engineered mouse model of lung adenocarcinoma, researchers observed that both physiological perturbation (jet lag) and genetic mutation of core circadian clock components led to decreased survival and promoted tumor growth and progression. Notably, circadian genes Per2 and Bmal1 demonstrated cell-autonomous tumor suppressive roles, influencing both transformation and lung tumor progression. Importantly, the loss of central clock components resulted in increased expression of c-Myc, heightened proliferation, and metabolic dysregulation. These findings underscore the critical role of metabolism alterations in driving cancer progression under conditions of circadian rhythm disruption [50]. Conversely, targeting metabolic pathways, such as the inhibition of glycolysis or the activation of oxidative phosphorylation, can promote apoptosis and enhance the efficacy of cancer therapy [48,51,52]. In addition to glucose metabolism, alterations in lipid metabolism have been implicated in lung cancer progression and resistance to apoptosis [53,54]. Lipid metabolism reprogramming, including enhanced lipid synthesis and altered lipid composition, promotes cancer cell survival and resistance to apoptosis [52]. Targeting lipid metabolism pathways, such as fatty acid synthesis or lipid droplet formation, represents a promising strategy to induce programmed cell death and overcome therapeutic resistance in lung cancer [55].

Immunonutritional interventions have emerged as promising strategies for modulating metabolic pathways and enhancing PCD in lung cancer [56]. These interventions utilize specific nutrients or dietary components with immunonutritional properties to exert therapeutic effects [57]. Omega-3 polyunsaturated fatty acids (PUFAs) have been extensively studied for their potential anti-cancer properties in lung cancer [58,59,60,61,62]. Preclinical studies have demonstrated the ability of omega-3 PUFAs to modulate lipid metabolism and promote apoptosis in lung cancer cells. Serini et al. [59] elucidated the mechanism by which omega-3 PUFAs induce apoptosis in lung cancer cells, showing the upregulation of pro-apoptotic proteins and inhibition of survival signaling pathways.

Additionally, omega-3 PUFAs have been found to suppress the expression of lipogenic genes, thereby impairing cancer cell growth and proliferation. Furthermore, clinical studies have provided evidence supporting the role of omega-3 PUFAs in lung cancer therapy. Murphy et al. [63] conducted a randomized controlled trial investigating the effects of omega-3 PUFA supplementation in patients with lung cancer undergoing chemotherapy. Their study demonstrated that supplementation with omega-3 PUFAs led to improved treatment outcomes, including increased apoptosis and decreased tumor growth compared with standard therapy alone. Moreover, omega-3 fatty acids have emerged as promising adjuncts to immunotherapy in lung cancer management because of their immunomodulatory properties and ability to induce apoptosis in tumor cells. Several studies have investigated the synergistic effects of omega-3 supplementation with immunotherapeutic agents, particularly immune checkpoint inhibitors, in both preclinical models and clinical trials [64,65,66,67].

In addition to omega-3 PUFAs, other immunonutritional interventions have shown promise in modulating metabolic pathways and enhancing programmed cell death in lung cancer [68,69,70]. For example, resveratrol, a natural compound with diverse health-promoting effects, exhibits promising anticancer properties in lung cancer [71,72]. It modulates enzymes involved in carcinogen metabolism and targets molecular pathways crucial for cancer development and progression [73]. Mechanistically, resveratrol induces apoptosis, cell cycle arrest, and senescence in lung cancer cells through various pathways including the TGF-β/Smad pathway, caspase activation, and p53 upregulation. Moreover, it synergizes with conventional cancer therapies like cisplatin and EGFR inhibitors, enhancing their efficacy and counteracting drug resistance. Resveratrol also influences microRNA expression, contributing to its antitumor effects [74]. Synthetic analogs of resveratrol, such as BCS and SS28, show potent growth inhibitory effects in lung cancer cells through mechanisms including cell cycle arrest and apoptosis induction [75,76]. These findings highlight the therapeutic potential of resveratrol and its analogs in lung cancer treatment, warranting further investigation into their clinical utility. *ω*-3 PUFAs are essential fatty acids that the human body cannot synthesize and are abundant in vegetable oils and fish fats. Docosahexaenoic acid (DHA) and eicosapentaenoic acid (EPA) are important components of *ω*-3 PUFAs, and they are also the most studied *ω*-3 PUFAs. *ω*-3 PUFAs have a certain role in preventing cardiovascular disease, adjusting inflammation, and improving nutritional status [77]. Ma’s research showed that *ω*-3 PUFAs downregulated CRP levels and reduced the duration of systemic inflammatory response syndrome (SIRS) [78]. However, studies by Lam CN and Carvalho TC showed that *ω*-3 PUFAs had no significant effect on nutritional improvement and inflammation regulation in patients with cancer [79,80]. Some randomized controlled trials have studied the therapeutic effect of *ω*-3 PUFAs on patients with lung cancer during radiotherapy and chemotherapy, but there is no corresponding meta-analysis study. This meta-analysis explores the efficacy of *ω*-3 PUFAs in patients with lung cancer undergoing radiotherapy and chemotherapy, so as to provide a reference for the treatment of patients with lung cancer.

## 7. Future Perspectives

The intricate interplay among programmed cell death (PCD), immunonutrition, and metabolism in the context of patients with lung presents a rich landscape for future investigations and therapeutic innovations. Firstly, further investigation into the molecular mechanisms underlying the crosstalk among immunonutrition, metabolism, and PCD in lung cancer is warranted. Elucidating the specific signaling pathways and regulatory networks involved in these interactions will provide valuable insights into the development of targeted therapies. Secondly, the development of novel immunonutritional interventions targeting specific metabolic pathways implicated in lung cancer metabolism holds promise for enhancing PCD and improving treatment outcomes. Research efforts aimed at identifying and validating potential therapeutic targets within these pathways will contribute to the development of personalized treatment strategies for lung cancer patients. Moreover, the integration of immunonutritional interventions with existing therapeutic modalities, such as chemotherapy, immunotherapy, and targeted therapy, represents a promising approach for enhancing treatment efficacy and overcoming therapeutic resistance in lung cancer. Clinical trials evaluating the synergistic effects of immunonutrition with standard-of-care treatments will be essential for translating preclinical findings into clinical practice.

## 8. Conclusions

Lung cancer poses a formidable challenge in oncology, necessitating innovative therapeutic strategies to enhance patient outcomes. PCD pathways, including apoptosis, autophagy, and programmed necrosis, play pivotal roles in regulating cancer cell fate and response to therapy. Dysregulation of these pathways contributes to tumor initiation, progression, and therapy resistance in lung cancer. Understanding the molecular mechanisms underlying PCD dysregulation in lung cancer provides valuable insights into potential therapeutic targets. Immunonutrition, involving specific nutrients to modulate immune function, shows promise as an adjunctive therapy in lung cancer management. Nutritional interventions targeting immunonutrition, such as omega-3 fatty acids, vitamins, minerals, and phytochemicals, modulate apoptosis and other forms of PCD in cancer cells. Omega-3 fatty acids, in particular, exhibit immunomodulatory properties and induce apoptosis in lung cancer cells, suggesting their potential as adjuvants to immunotherapy. Metabolic reprogramming, a hallmark of cancer cells including those in lung cancer, offers a promising therapeutic avenue by targeting metabolic pathways implicated in apoptosis regulation. Dysregulated glucose metabolism, lipid metabolism, and amino acid metabolism contribute to uncontrolled cell proliferation, survival, and resistance to apoptosis in lung cancer. Targeting key metabolic pathways involved in lung cancer metabolism, such as glycolysis, fatty acid synthesis and oxidation, and amino acid metabolism, emerges as a compelling therapeutic strategy. Clinical studies provide evidence supporting the role of immunonutritional interventions, such as omega-3 fatty acids, in enhancing programmed cell death and improving treatment outcomes in patients with lung cancer. Additionally, synthetic analogs of natural compounds, such as resveratrol, exhibit promising anticancer properties in lung cancer by modulating apoptotic signaling pathways and synergizing with conventional cancer therapies. In conclusion, the convergence of immunonutrition, metabolism, and programmed cell death pathways presents a fertile ground for therapeutic exploration in lung cancer. Further research is warranted to elucidate the specific molecular mechanisms underlying the interplay among these processes and to translate these findings into clinical practice.

Moreover, conducting well-designed clinical trials is essential to validate the efficacy and safety of immunonutritional and metabolic interventions in patients with lung cancer. Prospective studies with rigorous endpoints and biomarker assessments can provide valuable insights into their clinical utility and impact on treatment outcomes. Incorporating personalized medicine approaches is also critical for optimizing the use of these therapies in lung cancer. Integrating multi-omics data into treatment decision-making can identify patient subgroups most likely to benefit from specific interventions, thereby maximizing therapeutic efficacy and minimizing toxicity. Implementing translational research strategies, such as developing relevant preclinical models and establishing collaborative research networks, can expedite the clinical translation of promising therapies. Additionally, longitudinal studies and biomarker development efforts are needed to understand the long-term effects of interventions and guide patient selection in clinical practice. By addressing these research priorities, we can advance our understanding of immunonutrition, metabolism, and PCD pathways in lung cancer and develop more effective therapeutic strategies to improve patient outcomes.

## Data Availability

Not applicable.

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
