# Peer review of "Immunonutrition, Metabolism, and Programmed Cell Death in Lung Cancer: Translating Bench to Bedside"

_biology, 2024, doi:10.3390/biology13060409_

Round 1
Reviewer 1 Report
Comments and Suggestions for Authors
Reviewer 2 Report
Comments and Suggestions for Authors
Please find attached file.

Round 2
Reviewer 1 Report
Comments and Suggestions for Authors
The manuscript titled "Enhancing Immunonutrition and Metabolic Targeting in Lung Cancer Treatment: Mechanistic Insights and Future Directions" has undergone significant revisions based on the reviewer's comments, aimed at improving the clarity, depth, and impact of the work. The authors have diligently addressed the reviewer's suggestions, resulting in a more comprehensive and balanced manuscript. Below is a detailed evaluation of the changes made and the strengths and weaknesses of the revised manuscript.
- The revisions made to the manuscript have significantly enhanced its quality, clarity, and impact. The authors have effectively addressed the reviewer's comments, resulting in a more comprehensive and scholarly work. The manuscript now offers valuable insights into the mechanistic underpinnings, clinical implications, and future directions of immunonutrition and metabolic targeting in lung cancer treatment. I recommend acceptance of the revised manuscript for publication, as it makes a substantial contribution to the existing literature in the field.